# Development of the Connecting Piece in ODF1-Deficient Mouse Spermatids

**DOI:** 10.3390/ijms231810280

**Published:** 2022-09-07

**Authors:** Sigrid Hoyer-Fender

**Affiliations:** Johann-Friedrich-Blumenbach-Institute of Zoology and Anthropology—Developmental Biology, GZMB, Ernst-Caspari-Haus, Justus-von-Liebig-Weg 11, Georg-August-Universität Göttingen, 37077 Göttingen, Germany; shoyer@gwdg.de

**Keywords:** spermiogenesis, HTCA, connecting piece, centriole, capitulum, TEM

## Abstract

ODF1 is a major protein of the accessory fibres of the mammalian sperm tail. In addition, ODF1 is found in the connecting piece, a complex structure located at the posterior end of the nucleus that connects the sperm head and tail. The tight coupling of the sperm head and tail is critical for the progressive motility of the sperm to reach the oocyte for fertilisation. The depletion of ODF1 by homologous recombination in mice led to male infertility. Although sperm tails were present in the epididymis, no intact spermatozoa were found. Instead, the depletion of ODF1 resulted in sperm decapitation, suggesting that ODF1 is essential for the formation of the coupling apparatus and the tight linkage of the sperm head and tail. However, the development of the linkage complex in the absence of ODF1 has never been investigated. Here, I analysed the fine structure of the developing connecting piece by transmission electron microscopy. I show that the connecting piece develops as in wild-type spermatids. Structural abnormalities were not observed when ODF1 was absent. Thus, ODF1 is dispensable for the development of the connecting piece. However, the decapitation of ODF1-deficient spermatozoa indicates that the heads and tails of the spermatozoa are not linked, so that they separate when force is applied.

## 1. Introduction

Spermatozoa are generated by the cytodifferentiation of haploid spermatids, the products of the second meiotic division in the germinal epithelium of the testis in a process known as spermiogenesis [1,2]. The dramatic remodelling of the initially spherical spermatids includes nuclear condensation and elongation, as well as the formation of the acrosome, which is anteriorly located to the nucleus, and the sperm tail opposite it. Finally, the mature spermatozoa, which have now adopted their species-specific morphology, are released into the epididymis, where they acquire their final fertilising ability [3].

The sperm flagellum develops from the centriole pair that migrates from an anterior position toward the posterior region of the nucleus. The positions of the acrosome and the flagellum thus establish the bipolarity of the spermatozoon [4]. The centriole pair not only initiates flagellum formation but also forms a complex structure that links the sperm tail to the nucleus in a nuclear depression called the implantation fossa [4]. This linkage complex, which is known as the connecting piece, tightly couples the head to the tail [5]. This tight coupling is crucial for the progressive motility of the sperm and the transfer of the sperm nucleus to the oocyte at fertilisation [6].

The formation of the connecting piece and the sperm flagellum is initiated in each case by one of the two paired centrioles. While the distal centriole, which is oriented perpendicular to the cell membrane, assembles the flagellum, the connecting piece is assembled by the proximal centriole [7,8]. The first sign of the development of the connecting piece is the accumulation of dense material on the wall of the proximal centriole, which is close to the nuclear membrane and perpendicular to the longitudinal axis of the sperm flagellum, and on the wall of the distal centriole. The dense material progressively accumulates, eventually forming capitulum and striated columns. As the pair of centrioles relocates toward the nuclear membrane and eventually positions the proximal centriole in the implantation fossa, a local thickening forms on the nuclear membrane in the depression that is the basal plate. The basal plate and capitulum are close together but separated by a narrow space traversed by fine filaments [4]. These filaments, as well as direct contacts between basal plate and capitulum, most likely constitute the head-to-tail coupling apparatus (HTCA).

The striated or segmented columns fuse with the outer dense fibres (ODFs) of the sperm tail, which are located at the outer side of each microtubule doublet along the axoneme. The ODFs are prominent fibres that stiffen the sperm tail, protect it from shear forces, and promote its elastic recoil [9,10]. The main ODF proteins, especially ODF1/HSPB10 and ODF2, are also components of the connecting piece and localise in the basal plate, capitulum, and striated columns [11,12,13]. ODF1 is essential for the tight coupling of the sperm head to the tail [14]. The depletion of ODF1 by homologous recombination in mice causes male infertility due to sperm decapitation. Intact spermatozoa are completely missing in the epididymides of *Odf1*-deficient mice [14]. ODF1 is therefore an essential protein of the linkage complex, located in the connecting piece and mediating the attachment to the nucleus. Although the decapitation of *Odf1*-deficient sperm was discovered as a cause of male infertility, the development of the HTCA in the absence of ODF1 has not yet been studied. Here, I show by transmission electron microscopic studies that the HTCA in *Odf1*-deficient spermatids develops like that of wild-type spermatids. Obvious morphological anomalies were not detected. The failure of the head-to-tail coupling thus emanates from forces acting on the sperm, such as the shear forces acting during the transit into the epididymis.

## 2. Results

*Odf1*-deficient mice were generated by homologous recombination, replacing exon 1 and the translational start site with the neomycin cassette (Figure 1A). Correct integration of the neomycin coding sequence was verified by genotyping. Heterozygous males and females were fertile. The mating of heterozygous mice resulted in the birth of offspring in the expected Mendelian ratio of either wild-type, heterozygous, or homozygous *Odf1*-deficient animals. *Odf1*-deficiency in either the heterozygous or homozygous condition was verified by the amplification of exon 1 of the *Odf1* gene and the neomycin sequence integrated into exon 1 (Figure 1B). Homozygous animals were identified by the amplification of the neomycin sequence and the failure to amplify exon 1 of *Odf1* (Figure 1B).

Homozygous ODF1-deficient male mice were sterile, as confirmed by mating with wild-type females for several months without giving birth to offspring, while their heterozygous *Odf1^+/−^* male littermates were fertile and generally gave birth to pups after about 4 weeks of permanent mating with wild-type females. To clarify male infertility, testicular development was further investigated by histological inspection and gene expression analyses. The expression of marker genes for spermatogenic progression did not differ in either ODF1-deficient homozygous, *Odf1^+/−^*-heterozygous, or wild-type mice, indicating a normal progression of spermatogenesis even in the absence of ODF1 [14]. Furthermore, histological analyses of testicular sections over the course of testicular development showed the normal progression of spermatogenesis up to the stage of elongating spermatids, suggesting that ODF1 is not essential for either nuclear remodelling or acrosome biogenesis [14,15]. A disturbed seminiferous epithelium with displaced germ cells in the lumen and multinucleated cells was found only in testicular sections of adult mice with ODF1 deficiency. Additionally, mature spermatozoa were hardly found [14]. Furthermore, intact spermatozoa were absent from the caput epididymis of *Odf1*-deficient mice, although sperm tails were present [14]. In contrast, sperm heads and tails were clearly visible in the caput epididymis of wild-type mice. Examination of the testicular and epididymal sections by transmission electron microscopy revealed disturbed organisation of the mitochondrial sheath in the midpiece of the sperm flagellum and of the orderly arrangement of the outer dense fibres when ODF1 was absent [14]. The disturbance of the orderly end-to-end alignment of the elongated mitochondria that form helices in wild-type sperm is shown in Figure 2k (arrows). Although ODF1 was absent, the sperm tail developed with axoneme and outer dense fibres (Figure 2k,k’). Intact ODF1-deficient epididymal spermatozoa were not found. Instead, only acephalic sperm tails were present, showing some motility with greatly altered parameters, such as reduced velocity [14]. Remarkably, a reduction in ODF1 in the heterozygous condition resulted in reduced sperm motility without affecting fertility [14]. Heterozygous *Odf1^+/−^* sperm had a normal structure but showed an increase in the distance between the nuclear membrane and capitulum, indicating a loosening of the coupling between the sperm head and tail [16]. To clarify whether the development of the HTCA is affected when ODF1 is absent, I searched for the HTCA in the testicular sections of two randomly selected adult male mice using transmission electron microscopy. The developmental stages of the HTCA were only found in the testicular sections of one mouse, which was 6 months old when it was sacrificed.

The HTCA develops from the proximal centriole of the centriolar pair. The first sign of its development is the accumulation of dense material at the wall of the centriole while the pair of centrioles is positioned toward the posterior nuclear membrane. At this early stage, a slight nuclear depression was visible in the wild-type spermatid (Figure 2a,a’) but not at all in the *Odf1*-deficient spermatid (Figure 2b,b’). Furthermore, structural changes at the nuclear membrane were not visible. The nuclear depression in the wild-type spermatid indicated a more advanced stage in the development of the connecting piece but, most likely, not a substantial difference in HTCA development between wild-type and *Odf1*-deficient spermatids. As the centriole pair relocates toward the nuclear membrane, the nuclear depression becomes more pronounced, and a local thickening at the nuclear envelope in the indentation appears, forming the basal plate (bp). The nuclear indentation with the basal plate was visible in both wild-type and *Odf1*-deficient spermatids (Figure 2c,c’,d,d’, arrows). In addition, dense material accumulated on the wall of the juxta-nuclear proximal centriole that formed the anlage of the capitulum, which was positioned between the proximal centriole and the bp. In the next phase of the formation of the junction, the bp and capitulum were well-developed, and the segmented columns beneath the centrioles were visible (Figure 2e,e’,f,f’; segmented columns indicated by arrows). In both wild-type and ODF1-deficient spermatids, the basal plate formed a local thickening on the outer side of the nuclear envelope that lined the contours of the nuclear indentation (Figure 2e’,f’). A well-developed capitulum was present between the basal plate and the proximal centriole, following the contours of the nuclear indentation. The proximal centrioles of the wild-type and ODF1-deficient spermatids are shown here in cross-sections. In contrast to the cross-sections of the proximal centrioles, which are shown in Figure 2e,e’,f,f’, the longitudinal sections of the proximal centriole are shown in Figure 2g,h. The implantation fossa, basal plate, and capitulum were well-developed. In the wild-type spermatid, the sperm tail extended from the distal centriole, which is shown here in a longitudinal section (Figure 2g,g’). At the distal end of the proximal centriole, an extension named the adjunct appeared, which was barely visible in the wild-type (Figure 2g’, arrow) but distinct in the *Odf1*-deficient (Figure 2h’, arrow) spermatids. The adjunct is a temporary structure and disappears in the late stages of spermiogenesis. The distinctness of the adjunct in the *Odf1*-deficient spermatid was either due to the randomly more suitable cutting plane or a slightly different developmental stage, but it most likely does not reflect a significant difference in HTCA development. A cross-section through the proximal centriole together with a longitudinal section through the sperm tail in elongating *Odf1^−/−^* spermatids is shown in Figure 2i,i’. The basal plate (bp), capitulum, and segmented columns are clearly visible. The elongated spermatids show a well-developed connecting piece (Figure 2k,k’). Beyond that, I observed no fully developed and intact spermatid stages in the testis. In the normal condition, testicular spermatozoa are eventually released into the epididymis where they finally mature. However, I did not find spermatozoa with intact HTCA and head-to-tail linkage in the epididymides of *Odf1*-deficient mice, although they were full of sperm tails. When comparing the structure of the coupling apparatus of elongated *Odf1^−/−^* spermatids in the testis with that of *Odf1^+/^*^+^ spermatozoa in the epididymis, no obvious differences were found (Figure 2j,j’,k,k’). In both cases, a distinct basal plate lining the nuclear envelope in the implantation fossa, the opposite capitulum following the contours of the implantation fossa, and segmented columns were present. Fine filaments spanning the gap between the basal plate and capitulum, first described by Fawcett and Phillips [7], were particularly conspicuous in *Odf1*-deficient elongated spermatids (Figure 2k’). The observation of these filaments in the absence of ODF1 suggests that ODF1 is dispensable for their formation.

## 3. Discussion

ODF1 is a major protein of the outer dense fibres of the mammalian sperm tail. The ODFs are important for the stability and elasticity of the sperm tail but are not involved in active motility [9,10]. They are important for male fertility because tail abnormalities caused by impaired development of the ODFs are often found in infertile men [18]. The ODFs are composed of a couple of different proteins, among them are the major proteins ODF1 and ODF2 [11,12,13,19]. ODF1 is a small protein with a molecular mass of ~27 kDa, a high content of cysteine, and weak self-association [20]. The presence of a conserved α-crystallin domain indicated that ODF1 belongs to the superfamily of small heat shock proteins, which caused its renaming into HSPB10; thus, it may act as a molecular chaperone [21,22]. ODF1 deficiency in male mice caused sperm decapitation and infertility [14]. However, I observed no obvious structural anomalies in the development of the HTCA when ODF1 was missing. The regular structure of the connecting piece in spermatids and the absence of intact spermatozoa in the epididymis of *Odf1*-deficient mice indicate that the head and tail are not correctly linked. Therefore, decapitation easily occurs when any force is applied, such as shear forces during the transport from the testis to the epididymis. While the heavy sperm heads remain in the testis, the sperm tails are easily transported into the epididymis. A reduction in ODF1 as in the heterozygous condition moderately affected sperm parameters, especially their motility and the strength of the head-to-tail coupling [14,16]. Recently, ODF1 deficiency was identified as a factor in idiopathic male infertility [23]. The sperm of infertile men with severely reduced ODF1 levels are easily decapitated during mild stress treatment, indicating weakened head–tail attachment. In addition, sperm with reduced ODF1 levels exhibit abnormalities of the HTCA, such as malformed or reduced basal plate size, the presence of granular material near the connecting piece, or poorly developed or even absent striated columns [23]. Overall, ODF1 is also important in humans for the tight linkage of the sperm head to the tail [23]. In our study, no obvious differences in the development of the HTCA were found between ODF1-deficient and wild-type mouse spermatids. Although subtle differences might be hidden that could only be detected by morphometric analyses, this endeavour requires numerous micrographs of the same developmental stage and sectional plane, which could not be obtained.

A few other proteins were identified as essential components of the linkage complex that cause male infertility and sperm decapitation when missing [8]. Here, those proteins that are related to ODF1, either through similar subcellular distributions in the male germ cells or through interaction with ODF1, are of particular interest. ODF1 is located in the ODFs, segmented columns, capitulum, and basal plate; these locations are also shared by ODF2 [13]. ODF2-deficient mice are embryonically lethal, but heterozygous male mice are infertile due to neck-midpiece breakage of sperm [24]. ODF1 and ODF2 not only share the same location at similar substructures but also interact in a yeast two-hybrid assay [25]. Both proteins, additionally, interact with CCDC42, a coiled-coil protein essential for male fertility [26]. The targeted deletion of CCDC42 in mice caused multiple malformations of spermatids, such as the dislocation of the HTCA from the implantation fossa [27]. The neck region and the ODFs share several proteins. In addition to ODF1 and ODF2, ODF3, which is also named polyamine modulated factor 1 binding protein 1 (PMFBP1), is essential for the rigid connection between the head and tail. The deletion of *Odf3* in mice causes acephalic spermatozoa, and mutations in *ODF3* have been identified in patients with acephalic spermatozoa syndrome [28,29,30,31]. Several proteins interact with ODF1 in vitro, such as the serine-threonine kinase fused (Fu/Stk36) and the RING-finger protein OIP1 [32,33]. The inactivation of *Fu* in male germ cells led to infertility with numerous sperm abnormalities, such as decapitation, which is reminiscent of ODF1 depletion, while OIP1 has largely been uncharacterised.

All of these aforementioned proteins are cytoskeletal proteins that are found in the neck region of the HTCA. To firmly connect the sperm tail to the head, nuclear envelope proteins are required as part of the linking complex. In this context, proteins that bridge the nuclear envelope and link the nucleoskeleton to the cytoskeleton come into focus. In somatic cells, the LINC complex consists of SUN-domain proteins spanning the inner nuclear membrane and their interacting KASH-domain family proteins spanning the outer nuclear membrane and interacting with the cytoskeleton [34]. Of particular interest are the three testis-specific SUN-domain proteins, which have a specific subcellular distribution [35]. Although in vitro interaction between ODF1 and SPAG4/SUN4 was reported, it is not clear how a cytoskeletal protein can interact with a protein of the inner nuclear membrane in vivo [36]. In addition, SPAG4, as part of the linkage complex, has now been largely excluded [17,37,38]. In contrast, SPAG4L/SUN5 is essential for HTCA formation, as demonstrated by the *Sun5* deletion in mice and SUN5 mutations in infertile men with acephalic spermatozoa syndrome [39,40,41,42]. SUN5 interacts with Nesprin 3, which is essential for the head-to-tail linkage [43]. The interaction between SUN5 and Nesprin 3 thus bridges the nuclear envelope of male germ cells. Nesprin 3 spans the outer nuclear membrane and most likely interacts with cytoskeletal proteins, as in somatic cells. However, the cytoskeletal binding partners of Nesprin 3 in male germ cells have yet to be identified. In this regard, the focus should be on the proteins mentioned above, especially ODF1, ODF2, and ODF3.

## 4. Materials and Methods

### 4.1. Ethics Statement

Mouse experiments were reviewed and approved by the local ethics commission. The Institute of Human Genetics (University Medical Center of Göttingen) holds the licence for animal experiments (accession number AZ.33.11.42502/01-53.05). The guidelines of the German Animal Welfare Act (German Ministry of Agriculture, Health and Economic Cooperation) were strictly followed in all aspects of mouse work.

### 4.2. Generation of Odf1-Knockout Mice

*Odf1*-deficient mice were generated by homologous recombination, as described in [14,16]. In brief, the targeting vector was assembled from a genomic *Odf1* cosmid clone (121J1787Q3; 129/Ola; Resource Centre of the German Human Genome Project at the Max-Planck-Institute for Molecular Genetics, Berlin, Germany) and consisted of ~3 kb of the *Odf1* upstream genomic region and ~3.8 kb of the intronic sequences, both interrupted by the neomycin gene in pPNT-M1 [44]. The targeting construct was linearised and electroporated into ES cell line R1 (provided by A. Nagy, Toronto, ON, Canada). The homologous recombined ES cells were isolated by G418 (350 µg/mL) and ganciclovir (2µmol/L) selection and microinjected into 3.5-day-old embryos of the C57BL/6J strain [45]. The *Odf1*-knockout strain was established by mating with C57BL/6J females. The *Odf1* gene consists of two exons with the translational start site in exon 1. In the recombined allele, exon 1 was replaced by the *neomycin* coding sequence, resulting in a complete knockout of *Odf1* in the homozygous condition [14].

### 4.3. Genotyping

Genomic DNA was extracted from tail-tip biopsies using the Viagen DirectPCR-Tail (Viagen Biotech, Los Angeles, CA, USA) and Proteinase K digestion overnight, followed by heat inactivation. Genotyping was performed by PCR on genomic DNA using the primer pairs Odf1-I (ATCAACTCTGCCTGAGAC)/Neomy (CCTTCTATCGCCTCCTTGACG) for detection of the neomycin cassette in the recombined allele and Odf1-I/Odf1-N (GAGCTCAAGCTTTGGCCGCACTGAGTTGTC) for detection of the wild-type allele.

### 4.4. Fertility Tests and Sperm Motility Analyses

For fertility tests, *Odf1*-deficient males were mated with wild-type females for at least 3 months.

Epididymal spermatozoa were isolated by dissection of the tissue in IVF medium (Medi-Cult, Jyllinge, Denmark), followed by incubation for 1.5 h at 37 °C. Sperm movement was quantified using the CEROS computer-assisted semen analysis system (version 10; Hamilton Thorne Research, Beverly, MA, USA), as described in [14,16].

### 4.5. Transmission Electron Microscopy (TEM)

Freshly isolated testes from adult mice were cut into small pieces and immediately fixed by immersion in 5% glutaraldehyde in 0.2 M phosphate buffer (pH 7.4) for 24 h, followed by washing in 0.2 M phosphate buffer (pH 7.4). For epoxy resin embedding, the samples were postfixed in 1% osmium tetroxide in aqua bidest, stained in half-saturated watery uranyl acetate, dehydrated in an ascending ethanol series, and finally embedded in Agar 100 (Agar Scientific Ltd., Stansted, UK). Ultrathin sections (~70 nm) were cut using an ultramicrotome and examined by TEM (Zeiss EM 900, Oberkochen, Germany). Digital images were captured with a slow-scan 2K CCD camera (TRS, Tröndle, Moorenweis, Germany) and processed using Adobe Photoshop CS5 (Adobe Inc., San Jose, CA, USA).

## Figures and Tables

**Figure 1 ijms-23-10280-f001:**
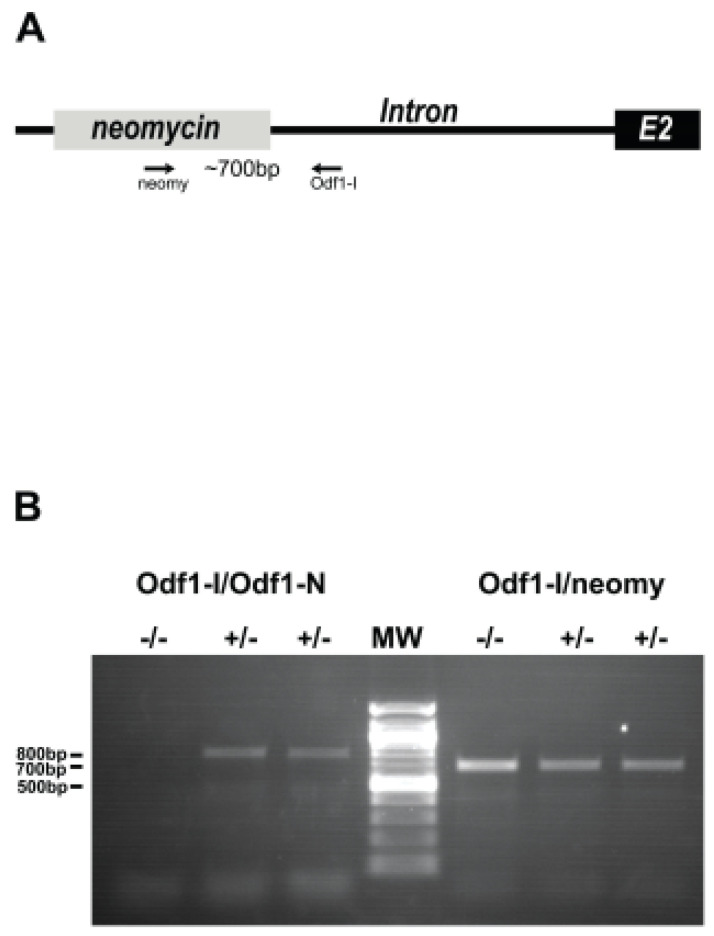
Genotyping of *Odf1*-deficient mice. (**A**) Replacement of exon 1 by neomycin. The positions of the primer pair for amplification of the insertion are indicated. A product of ~700 bp is expected. E2 = exon 2. (**B**) Genotyping of homozygous (−/−) and heterozygous (+/−) *Odf1*-deficient mice. Amplification of exon 1 of *Odf1* with the primer pair Odf1-I/Odf1-N and of the neomycin insertion with primer pair Odf1-I/neomy. The neomycin insertion was amplified in homozygous *Odf1*-deficient mice (−/−) but not exon 1 of *Odf1*. MW = 100 bp ladder.

**Figure 2 ijms-23-10280-f002:**
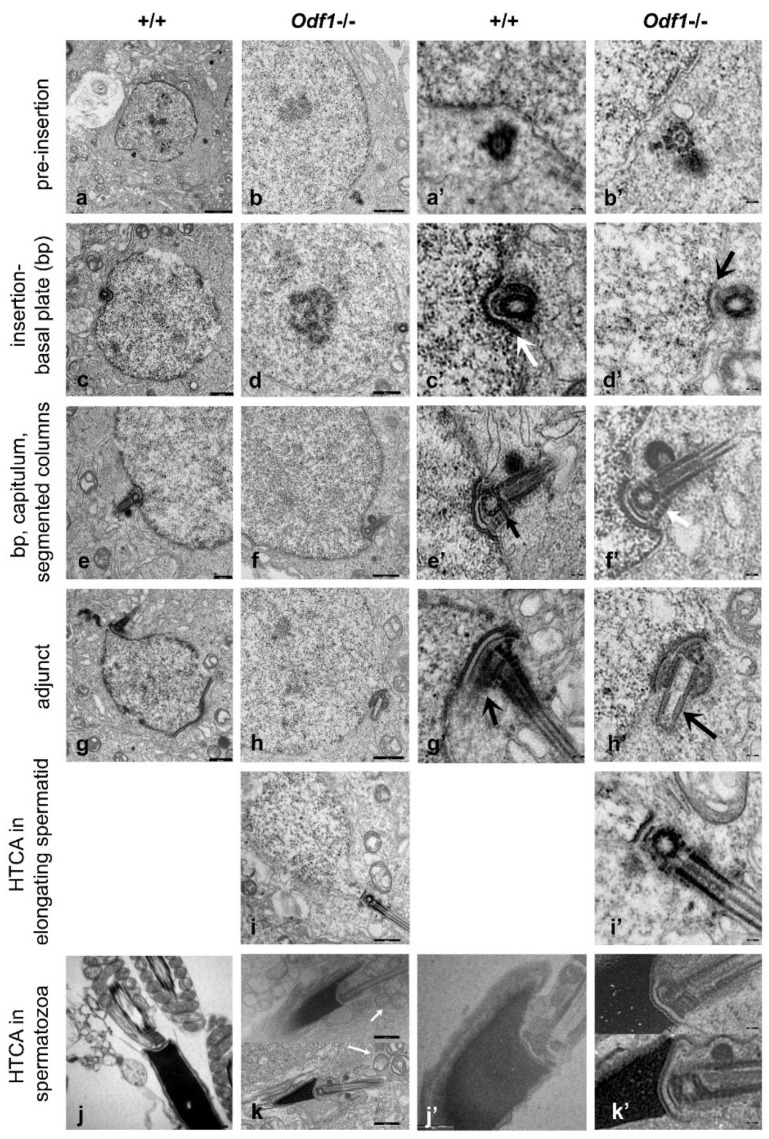
Transmission electron micrographs of the developing connecting piece in *Odf1*-deficient (−/−) and wild-type spermatids (+/+). First sign of the development of the connecting piece before insertion into the implantation fossa (**a**,**b**) and respective enlargements (**a’**,**b’**). The proximal centriole is inserted into the implantation fossa and the basal plate (bp) develops (**c**,**d**) and respective enlargements (**c****’**,**d****’**; arrows indicate bp). Basal plate (bp), capitulum, and segmented columns are developed (**e**,**f**) and respective enlargements (**e’**,**f’**; arrows indicate segmented columns). The adjunct is visible (**g**,**h**) and enlargements (**g’**,**h’**; the adjunct is indicated by arrows). The head-to-tail coupling apparatus (HTCA) in elongating spermatids in the absence of ODF1 (**i**,**i’**). Fully developed HTCA in elongated *Odf1*-deficient spermatozoa in the testis (**k**) and enlargement (**k’**); the arrows in k indicate the disorganised mitochondria, and in epididymal *Odf1^+/+^* spermatozoa (**j**,**j’**). (**j**) *Odf1^+/+^* spermatozoa of 129Sv background, (**j’**) *Odf1^+/+^- but Spag4^−/−^*-deficient spermatozoa (**j’**). Scale bars: 2500 nm (**a**), 1000 nm (**b**–**d**,**f**–**i**,**k,a’**–**i’**,**k’**), 500 nm (**e**,**j’**), 100 nm (**j**). (**g**,**g’**,**j**) Reprinted by permission from Springer Nature (license number 5358120337911): Springer Nature, Histochemistry and Cell Biology, Ultra-structure of the sperm head-to-tail linkage complex in the absence of the spermatid-specific LINC component SPAG4, Yang et al., 2018 [17].

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
