# Peer review of "Development of the Connecting Piece in ODF1-Deficient Mouse Spermatids"

_ijms, 2022, doi:10.3390/ijms231810280_

Round 1
Reviewer 1 Report
This is an interesting study to the field of reproductive biology and male fertility research. The author report generation of Odf1-deficient mice. Interestingly, homozygous male mice are infertile indicating an essential role of Odf1 in male fertility due to lack of spermatozoa with intact HTCA and head-to-tail linkage. The manuscript is well written and results are supported by experimental evidence. I have few minor points that could improve the manuscript.
Section 2.3: Please mention number of mice/sections used for TEM.
Please indicate age of mice used in the current analysis.
Lines 107-111, please rephrase and clarify this part.
Do detached sperm heads of Odf1-deficient mice do successful ICSI?
It would be great to include data on spatio-temporal localization of Odf1 in sperm cells during spermiogenesis and in the coupling apparatus.
CASA data on sperm motility of WT and Odf1-deficient mice may be useful to include.
What is the percentage of tails with sperm heads and tails only in cauda, caput and corpus epididymis?
Do Odf1-deficiency/mutations affect sperm motility/head-tail linkage and fertility in human?
Please clearly describe the assembly as well as attachment of the capitulum and basal plate to the nuclear envelope in the implantation fossa of both WT and Odf1-deficient spermatids.
Morphometric analysis may be required for figure 2 to resolve subtle differences in Odf1-deficient sperm head, tail and HTCA.
Does Odf1 have a role in acrosome biogenesis and nuclear remodeling?
Please discuss the reasons why in Odf1-deficient testes, the decapitated flagella were released while heads remained in the seminiferous tubules. In other words, what is the effect of spermatid elongation process on nuclear envelope-coupling apparatus interaction and the detachment of basal plate-capitulum-segmented column complex from nuclear envelope?
Author Response
Reviewer 1:
- Section 2.3: Please mention number of mice/sections used for TEM.
Reply: More than 20 homozygous Odf1-deficient male mice were analysed and used for histology, investigation of testicular development, gene expression analyses, and sperm motility analyses. Half of them were mated with wild-type females and all of them were infertile after several months of permanent mating. Testes and epididymides of 4 randomly selected mice were analysed by semi-thin sectioning and electron microscopy. Investigation of semi-thin sections revealed sperm tails in the epididymides, malformed sperm heads and immature germ cells but no normal sperm heads. Transmission electron microscopy of Odf1-deficient epididymides showed sperm tails with irregular mitochondrial arrangement but no sperm heads, whereas sperm heads and tails were present in the epididymides of wild-type mice. The HTCA was hardly detectable by transmission electron microscopy in testicular sections of Odf1-deficient mice. Therefore, we were happy to identify the developing HTCA at all in the absence of ODF1. Thus, HTCA development could be demonstrated only in testicular sections of one mouse.
The text was changed into (from line 78 on):
Homozygous ODF1-deficient male mice were sterile, as confirmed by mating with wild-type females for several months without giving birth to offspring, while their heterozygous Odf1+/--male littermates were fertile and generally gave birth to pups after about 4 weeks of permanent mating with wild-type females. To clarify male infertility, testicular development was further investigated by histological inspection and gene expression analyses. The expression of marker genes for spermatogenic progression did not differ in either ODF1-deficient homozygous, Odf1+/--heterozygous or wild-type mice, indicating normal progression of spermatogenesis even in the absence of ODF1 [14]. Furthermore, histological analyses of testicular sections over the time course of testicular development showed a normal progression of spermatogenesis up to the stage of elongating spermatids, suggesting that ODF1 is not essential for either nuclear remodelling or acrosome biogenesis [14, 15]. A disturbed seminiferous epithelium with displaced germ cells in the lumen and multinucleated cells was found only in testicular sections of adult mice with ODF1-deficiency. Additionally, mature spermatozoa are hardly found [14]. Furthermore, intact spermatozoa are absent from the caput epididymis of Odf1-deficient mice, although sperm tails were present [14]. In contrast, sperm heads and tails were clearly visible in the caput epididymis of wild-type mice. Examination of testicular and epididymal sections by transmission electron microscopy revealed disturbed organization of the mitochondrial sheath in the mid-piece of the sperm flagellum and of the orderly arrangement of the outer dense fibres when ODF1 is absent [14]. The disturbance of the orderly end-to-end alignment of the elongated mitochondria that form helices in wild-type sperm is shown in Fig. 2k (arrows). Although ODF1 is absent, the sperm tail developed with axoneme and outer dense fibres (Fig. 2k, k’). Intact ODF1-deficient epididymal spermatozoa were not found. Instead, only acephalic sperm tails were present, showing some motility with greatly altered parameters such as reduced velocity [14]. Remarkably, reduction of ODF1 in the heterozygous condition resulted in reduced sperm motility without affecting fertility [14]. Heterozygous Odf1+/--sperm had a normal structure, but showed an increase in the distance between the nuclear membrane and the capitulum, indicating a loosening of the coupling between sperm head and tail [16]. To clarify whether the development of the HTCA is affected when ODF1 is absent, we searched for the HTCA in testicular sections of two randomly selected adult male mice using transmission electron microscopy. Developmental stages of the HTCA were only found in the testicular sections of one mouse, which was 6-months old when it was sacrificed.
- Please indicate age of mice used in the current analysis.
Reply: information is now included (see above):
Developmental stages of the HTCA were only found in the testicular sections of one mouse, which was 6-months old when it was sacrificed.
- Lines 107-111, please rephrase and clarify this part.
Changed into:
The basal plate (bp), capitulum, and segmented columns are clearly visible. The elongated spermatids show a well-developed connecting piece (Fig. 2k, k’). Beyond that, no fully developed spermatid stages could be observed in the testis. In the normal condition, testicular spermatozoa are eventually released into the epididymis where they finally mature. However, spermatozoa with intact HTCA and head-to-tail linkage were never found in the epididymides of Odf1-deficient mice albeit full of sperm tails.
- Do detached sperm heads of Odf1-deficient mice do successful ICSI?
This is an interesting question regarding the importance of sperm centrioles for early development. However, this question might be more important for humans since human spermatozoa contain two remodelled centrioles whereas mouse sperm lack centrioles at all. Unfortunately, since sperm heads were not found in ODF1-deficient mice, ICSI was not possible to perform.
- It would be great to include data on spatio-temporal localization of Odf1 in sperm cells during spermiogenesis and in the coupling apparatus.
Reply: I am not able to provide these data because there are no live animals left, testicular sections have not been processed for immunoelectron microscopy, the anti-ODF1 antibody has not been tested for its suitability for immuno-EM and finally the difficulties in finding developing HTCA stages in testicular sections.
Immuno-EM has been performed as cited demonstrating the presence of ODF1 in the connecting piece (line 53). The text was changed into:
The main ODF proteins, especially ODF1/HSPB10 and ODF2, are also components of the connecting piece and localize in the basal plate, capitulum, and striated columns [11-13].
- CASA data on sperm motility of WT and Odf1-deficient mice may be useful to include.
Reply: CASA data have already been published as Supplementary Fig. 1 (S1) in [14].
But I add the following in the text (line 95):
Intact ODF1-deficient epididymal spermatozoa were not found. Instead, only acephalic sperm tails were present, showing some motility with greatly altered parameters such as reduced velocity [14].
- What is the percentage of tails with sperm heads and tails only in cauda, caput and corpus epididymis?
Reply:
We stated the following (line 89):
Furthermore, intact spermatozoa are absent from the caput epididymis of Odf1-deficient mice, although sperm tails were present [14].
Since we observed no intact spermatozoa in the caput (only acephalic tails) no intact spermatozoa would be found in corpus and cauda as well.
- Do Odf1-deficiency/mutations affect sperm motility/head-tail linkage and fertility in human?
Reply: yes!
The following text is now included (line 173):
Sperm of infertile men with severely reduced ODF1 levels are easily decapitated during mild stress treatment, indicating weakened head-tail attachment. In addition, sperm with reduced ODF1 levels exhibit abnormalities of the HTCA, such as malformed or reduced basal plate size, the presence of granular material near the connecting piece, or poorly developed or even absent striated columns [22]. Overall, ODF1 is also important in humans for the tight linkage of the sperm head to the tail [22].
- Please clearly describe the assembly as well as attachment of the capitulum and basal plate to the nuclear envelope in the implantation fossa of both WT and Odf1-deficient spermatids.
Reply: the following was included:
Line 116:
In both wild-type and ODF1-deficient spermatids, the basal plate forms a local thickening on the outer side of the nuclear envelope that lines the contours of the nuclear indentation (Fig. 2 e’, f’). A well-developed capitulum is present between the basal plate and the proximal centriole, following the contours of the nuclear indentation. The proximal centrioles of the wild-type and ODF1-deficient spermatids are shown here in cross-sections.
Line 135:
In both cases, a distinct basal plate lining the nuclear envelope in the implantation fossa, the opposite capitulum following the contours of the implantation fossa, and segmented columns are present. Fine filaments spanning the gap between basal plate and capitulum, first described by Fawcett and Phillips [7], are particularly conspicuous in Odf1-deficient elongated spermatids (Fig. 2 k’). The observation of these filaments in the absence of ODF1 suggests that ODF1 is dispensable for their formation.
- Morphometric analysis may be required for figure 2 to resolve subtle differences in Odf1-deficient sperm head, tail and HTCA.
Reply: text added (line 178):
In our study, no obvious differences in the development of the HTCA were found between ODF1-deficient and wild-type spermatids. Although subtle differences might be hidden that could only be detected by morphometric analyses, this endeavour requires numerous micrographs of the same developmental stage and sectional plane, which could not be obtained.
Note added:
Furthermore, I would mention that the micrographs of Hetherington et al., [22] are of low quality and have not been analysed morphometrically.
- Does Odf1 have a role in acrosome biogenesis and nuclear remodeling?
Reply: Spermatogenesis proceeded normally up to the elongated spermatid stage even in the absence of ODF1 indicating that ODF1 may not be involved in nuclear remodelling. Furthermore, the acrosome was detected in ODF1-deficient testicular sections as published in Tapia Contreras and Hoyer-Fender, 2020. Scientific Reports; https://doi.org/10.1038/s41598-020-71120-9.
We included the following (line 84):
Furthermore, histological analyses of testicular sections over the time course of testicular development showed a normal progression of spermatogenesis up to the stage of elongating spermatids, suggesting that ODF1 is not essential for either nuclear remodelling or acrosome biogenesis [14, 15].
- Please discuss the reasons why in Odf1-deficient testes, the decapitated flagella were released while heads remained in the seminiferous tubules. In other words, what is the effect of spermatid elongation process on nuclear envelope-coupling apparatus interaction and the detachment of basal plate-capitulum-segmented column complex from nuclear envelope?
Reply:
The most likely reason why decapitated flagella are found in the epididymis but not the sperm heads is that the connection between head and tail is disturbed when ODF1 is lacking. Albeit the HTCA developed normally without any obvious differences compared to the wild-type, the molecular linker is missing, which could not be detected visually. However, when shear forces act on these cells, such as during the transit from the testis into the epididymis, decapitation occurs. The heavy sperm heads would stay in the testis, whereas the tails are easily transported into the epididymis.
Since elongated spermatozoa with a regular HTCA are found in the testis, I do not think that the elongation process has any effect on the HTCA. I therefore omitted this discussion.
I add the following (line 167):
The regular structure of the connecting piece in spermatids and the absence of intact spermatozoa in the epididymis of Odf1-deficient mice indicate that head and tail are not correctly linked. Therefore, decapitation easily occurs when any force is applied, such as shear forces during the transport from the testis to the epididymis. While the heavy sperm heads remain in the testis, the sperm tails are easily transported into the epididymis.
Reviewer 2 Report
In this manuscript entitled “Development of the connecting piece in ODF1-deficient mouse spermatids”, the authors analyzed the detailed morphologies of the connecting piece with a TEM.
Figure 1 shows that the authors used heterozygous (+/-) mice as controls. If so, the heterozygous mice should also be used in Figure 2. Therefore, there are major flaws in the experimental design.
Comments:
1. The description of the method is insufficient. The authors should write in detail.
2. Are the properties (length, thickness, etc.) of the formed base plate the same as the control? The authors should write it in detail.
3. Fig 2 g’ and h’: Because of the poor quality of the micrograph of 'g', it is not clear whether each of the adducts indicated by the arrows shows the same position. The authors should present high-quality micrographs.
4. Fig 2 k and k’: Odf1-deficient male mice are infertile due to the detachment of the sperm head [14]. Why are sperm heads present and attached to the tail in these electron micrographs?
Author Response
Figure 1 shows that the authors used heterozygous (+/-) mice as controls. If so, the heterozygous mice should also be used in Figure 2. Therefore, there are major flaws in the experimental design.
Reply:
In this case, the heterozygous animals were used to demonstrate the plausibility of the genotyping results.
By investigating the HTCA of sperm of Odf1+/--heterozygous and wild-type mice we found no obvious differences. Subtle differences were discovered only by morphometric analyses, which demonstrated a loosened connection between sperm head and tail when ODF1 was reduced [ Yang K, Grzmil P, Meinhardt A & Hoyer-Fender S 2014 Haplo-Deficiency of ODF1/HSPB10 in Mouse Sperm Causes Relaxation of Head-to-Tail Linkage. Reproduction 148 499–506. doi:10.1530/REP-14-0370.]. Our data suggested that the development of the HTCA in Odf1+/--heterozygous germ cells is similar if not identical to that in wild-type germ cells. We expected nothing informative when investigating the HTCA in heterozygous germ cells, which led to our decision to omit the investigation of the HTCA in heterozygous germ cells.
Comments:
- The description of the method is insufficient. The authors should write in detail.
Reply: Has been improved as follows:
Generation of Odf1-knock out mice
Odf1-deficient mice were generated by homologous recombination as described in [14, 16]. In brief, the targeting vector was assembled from a genomic Odf1 cosmid clone (121J1787Q3; 129/Ola; Resource Centre of the German Human Genome Project at the Max-Planck-Institute for Molecular Genetics, Berlin, Germany), and consists of ~3kb of the Odf1 upstream genomic region and ~3.8kb of the intronic sequences, both interrupted by the neomycin gene in pPNT-M1 [44]. The targeting construct was linearized and electroporated into ES cell line R1 (provided by A. Nagy, Toronta, Ontario, Canada). Homologous recombined ES cells were isolated by G418 (350µg/ml) and ganciclovir (2µmol/liter) selection, and micro-injected into 3.5-day-old embryos of the C57BL/6J strain [45]. The Odf1-knock out strain was established by mating with C57BL/6J females….
Genotyping
Genomic DNA was extracted from tail-tip biopsies using Viagen DirectPCR-Tail (Viagen Biotech, Los Angeles, CA, USA) and Proteinase K digestion overnight, followed by heat inactivation. Genotyping was done by PCR on genomic DNA using the primer pairs Odf1-I…
Fertility tests and sperm motility analyses
For fertility test, Odf1-deficient males were mated with wild-type females for at least 3 months.
Epididymal spermatozoa were isolated by dissection of the tissue in IVF medium (Medi-Cult, Jyllinge, Denmark) followed by incubation for 1.5hrs at 37°C. Sperm movement was quantified using the CEROS computer-assisted semen analysis system (version 10; Hamilton Thorne Research, Beverly, MA, USA) as described in [14, 16].
Transmission Electron microscopy (TEM)
Freshly isolated testis was cut in pieces and immediately fixed by immersion in 5% glutaraldehyde in 0.2 M phosphate buffer (pH 7.4) for 24hrs. The tissue was then washed in 0.2 M phosphate buffer (pH 7.4), followed by post-fixation in 2% osmium tetroxide, and finally embedded in epoxy (Epon) resin. Specimens were first sectioned semi-thin and inspected. Selected areas were then sectioned ultra-thin and examined by transmission electron microscopy.
- Are the properties (length, thickness, etc.) of the formed base plate the same as the control? The authors should write it in detail.
Reply: text added (line 178):
In our study, no obvious differences in the development of the HTCA were found between ODF1-deficient and wild-type spermatids. Although subtle differences might be hidden that could only be detected by morphometric analyses, this endeavour requires numerous micrographs of the same developmental stage and sectional plane, which could not be obtained.
- Fig 2 g’ and h’: Because of the poor quality of the micrograph of ‘g’, it is not clear whether each of the adducts indicated by the arrows shows the same position. The authors should present high-quality micrographs.
Reply:
I have to admit that I cannot provide a better picture (Fig. 2g, g’) which is comparable
in developmental stage and sectioning with that of the Odf1-/--deficient spermatid.
I have already gone into the subtle differences in the text:
(line 123):
At the distal end of the proximal centriole an extension appears, named the adjunct, which is
barely visible in the wild-type (Fig. 2g’, arrow) but distinct in the Odf1-deficient spermatid (Fig.
2h’, arrow). The adjunct is a temporary structure and disappears in the late stages of
spermiogenesis. The distinctness of the adjunct in the Odf1-deficient spermatid is either due
to the randomly more suitable cutting plane or to a slightly different developmental stage,
but most likely does not reflect a significant difference in HTCA development.
- Fig 2 k and k’: Odf1-deficient male mice are infertile due to the detachment of the
sperm head [14]. Why are sperm heads present and attached to the tail in these
electron micrographs?
Reply: In Fig. 2k, k’ elongated spermatids are shown, but no fully developed sperm. These are
the most advanced stages observed in ODF1-deficient testis. Intact sperm could not be found
in either the testis or the epididymis. For comparison, we included pictures of Odf1+/+-
epididymal sperm (Fig. 2j, j’).
Round 2
Reviewer 2 Report
- The description of the method is insufficient. The authors should write in detail.
Transmission Electron microscopy (TEM)
Freshly isolated testis was cut in pieces and immediately fixed by immersion in 5% glutaraldehyde in 0.2 M phosphate buffer (pH 7.4) for 24hrs. The tissue was then washed in 0.2 M phosphate buffer (pH 7.4), followed by post-fixation in 2% osmium tetroxide, and finally embedded in epoxy (Epon) resin. Specimens were first sectioned semi-thin and inspected. Selected areas were then sectioned ultra-thin and examined by transmission electron microscopy.
Comments:
The manuscript does not detail the methods, even though the TEM results are the main focus. Did the authors observe with osmium staining only? How thick was the cut? The authors should address on a detailed TEM method description, including observation conditions, etc.
Author Response
Transmission Electron microscopy (TEM)
Freshly isolated testis from adult mice was cut in small pieces and immediately fixed by immersion in 5% glutaraldehyde in 0.2 M phosphate buffer (pH 7.4) for 24hrs, followed by washing in 0.2 M phosphate buffer (pH 7.4). For epoxy resin embedding, samples were postfixed in 1% osmium tetroxide in aqua bidest, stained in half-saturated watery uranyl acetate, dehydrated in an ascending ethanol series and finally embedded in Agar 100 (Agar scientific Ltd., Stansted, UK). Ultrathin sections (~ 70 nm) were cut using an ultramicrotome and examined by TEM (Zeiss EM 900, Oberkochen, Germany). Digital images were captured with a slow-scan 2K CCD camera (TRS, Tröndle, Moorenweis, Germany) and processed using Adobe Photoshop.